# Toward an Experience-Based Model of Recovery and Recovery-Oriented Practice in Mental Health and Substance Use Care: An Integration of the Findings from a Set of Meta-Syntheses

**DOI:** 10.3390/ijerph20166607

**Published:** 2023-08-18

**Authors:** Trude Klevan, Mona Sommer, Marit Borg, Bengt Karlsson, Rolf Sundet, Hesook Suzie Kim

**Affiliations:** Center for Mental Health and Substance Abuse, Department of Health, Social and Welfare Studies, Faculty of Health and Social Sciences, University of South-Eastern Norway (USN), 3040 Drammen, Norway; marit.borg@usn.no (M.B.); bengt.karlsson@usn.no (B.K.); rolf.sundet@usn.no (R.S.); hsuziekim@comcast.net (H.S.K.)

**Keywords:** recovery, mental health and substance use, meta-synthesis, experience-based model

## Abstract

A model of recovery and recovery-oriented practice has been developed based on three previously published meta-syntheses of experiences and processes of mental health and substance use recovery. The model integrates the findings of these three meta-syntheses into three components: experiences of recovery, processes of recovery-oriented practice, and social and material capital. The experiences of recovery involve being, doing, and accessing and are viewed as embedded in the processes of recovery. The processes of recovery-oriented practice aim to mobilize and apply various forms of capital to support the recovery journey. Social and material capital, in turn, constitute the context in which recovery occurs and requires mobilization for the individual and the service system. The model is grounded in the principles of well-being, person-centeredness, embedding, self-determination, and the interdependency of human living. The model is both descriptive and explanatory, as it depicts the experiential and processual aspects of recovery and recovery-oriented practice and their interrelationships. The model as a framework needs to be elaborated further through application in practice and research, especially for understanding how experiences, processes and practices interact over time, and how they are affected by access to material and social capital.

## 1. Introduction

In enhancing knowledge of mental health and substance use recovery and recovery-oriented services, the participation of individuals with lived experience is crucial. The World Health Organization (2022) [1] states that the right to participate is an essential feature of the right to the highest attainable standard of health. Lived experiences provide powerful expertise that is important in shaping policies and informing those in power. Recognizing lived experience as a valuable source of knowledge in determining how health can be understood, what role contextual factors play and what kinds of help are considered useful, is important in what is known as recovery in the field of mental health and substance use. The concept of recovery remains multifaceted and contested. In this paper, we position recovery as stemming from the civil rights movements that began in the 1960s, as a response to stigma and suppression and based on ideas of human rights and empowerment. We thereby understand recovery as subjective experiences, encompassing personal, social, and contextual processes. Thus, at its heart, recovery is an experiential concept and what are often described as recovery-oriented services need to be informed by experiential knowledge. Given its experiential, subjective and contextual nature, recovery is not a theoretical model that can be implemented in services. However, we suggest that a descriptive and explanatory model of recovery, developed from experience-based knowledge, can be a useful tool in exploring and understanding recovery as complex, contextual and collaborative processes.

This paper presents an experience-based model of recovery and recovery-oriented practice developed by integrating the findings from three meta-syntheses [2,3,4]. These meta-syntheses were based on the empirical publications of the research team of the Center for Mental Health and Substance Abuse of the University of South-Eastern Norway, published in the period 2005–2020. Altogether, 74 empirical papers were included in the three previously published meta-syntheses [2,3,4]. The results of these three meta-syntheses are expressed through the following overarching themes: (1) the experiential nature of recovery, (2) recovery processes, and (3) recovery-oriented practice. The results are based on lived experiences narrated by persons in recovery, by their families, or by professional providers for persons in community mental health and substance use care. A model of recovery and recovery-oriented practice has thus been developed, specifying experientially rich knowledge and elaborating the key features of recovery and recovery-oriented practice as an integrated framework. This model is presented by reflecting on the principles of wellbeing, agency/person-centeredness, and the interdependence/social nature of human life, and proposes interrelationships between experiences of recovery, recovery processes, and recovery-oriented practice as a comprehensive way of understanding these experiences.

## 2. Background and Foundations

Although there are still controversial views regarding the concept of recovery and its application in the context of mental health and substance use (MHSU) care, specifically in terms of the dichotomies of clinical versus personal and process versus outcome, the definition of recovery in the MHSU context advanced by SAMHSA is generally accepted. It states that “recovery is a process of change through which individuals improve their health and wellness, live a self-directed life, and strive to reach their full potential” [5] (p. 3). Recovery encompasses non-linear, personal, social, and contextual processes through which people strive to deal with their MHSU difficulties and associated life experiences in order to lead fulfilling lives. The systematic review of the reviews of the literature on mental health recovery by Dell, Long and Mancini (6) extracted five key themes of recovery that support this view, which are: (a) recovery is a process of overcoming despair to realize a positive sense of self and wellbeing; (b) recovery requires an environment marked by safety, security, and access to basic resources; (c) recovery involves the exercise of autonomy, control, and personal responsibility; (d) recovery is dependent on a sense of belonging, meaning, and purpose derived through social support, connection, and meaningful activities and roles; and (e) recovery is dependent on acceptance of the illness as a part of oneself and the development of insight into how to manage the illness and maintain wellbeing. While recovery is essentially a personal experience, it is at the same time critically embedded in the interdependence in human life of our relationships with others, social and environmental contexts, and various forms of material and non-material resources [6,7]. The term relational recovery suggests that recovery is always relational, taking place in interactions and relationships with people and things [7]. Furthermore, recovery is connected to human rights and citizenship, emphasizing how social conditions, inequity and marginalization in relation to mental health and substance use issues affect people’s lives and possibilities [8,9]. Rowe and Davidson [10] suggest that recovery emerges through citizenship, rather than citizenship being a prerequisite for recovery. Recovering citizenship may be achieved through an orientation towards the five Rs: rights, responsibilities, roles, resources and relationships. According to these authors, the availability of the five Rs for all citizens is an important characteristic of a democratic society.

Recovery-oriented practice in MHSU care encompasses person-centered processes such as acceptance and insight, autonomy and control as the basis for the recovery process of self-actualization, transformation from a negative to a positive sense of self, and provider-oriented processes of support and resource mobilization to help people to pursue their subjective recovery processes and handle difficulties that may occur. During recent decades, many versions of recovery-oriented practice models have been developed and applied in clinical practice. Several systematic reviews focus on the nature and characteristics of recovery-oriented practice, beginning with the study by Le Boutillier and colleagues (2011) [11] with an international focus, which identified 16 guidance dimensions for recovery-oriented mental health practice that supports self-advocacy, aimed at both mental health services/practitioners and service users, which culminated in a conceptual framework of four dimensions of recovery-oriented practice as (a) promoting citizenship, (b) organizational commitment, (c) supporting personally defined recovery, and (d) a working relationship. Furthermore, the CHIME framework has been widely referenced and used in the understanding of recovery and the development of recovery-oriented services. This framework focuses on personal recovery and was developed through a systematic review and narrative synthesis of recovery. It consists of three superordinate categories of recovery, namely, characteristics of the recovery journey, recovery processes, and recovery stages. The framework identifies five important recovery processes, making up the acronym CHIME: connectedness; hope and optimism; identity, meaning and purpose; and empowerment [12,13]. Stuart and colleagues [14] later suggested that a D for “difficulties” might be added to the framework, and that a CHIME-D framework would involve an important recognition of how MHSU distress often encompasses difficulties at personal and structural levels. The authors suggest that acknowledging difficulties might be a small step towards challenging a person-oriented understanding of recovery, recognizing how structural deficits might cause distress and prevent recovery. While these are principal elements desirable for recovery-oriented practice, the perceptions of practitioners regarding recovery-oriented practice have been found to be a mixture of the perspectives from clinical recovery from the deficit perspective; personal recovery as holistic, individualistic, and citizenship oriented; and service-defined recovery [11]. On the other hand, Chester et al. [15] found two main themes in delivering recovery-oriented practice for severe mental illness to be (a) alleviating stigma by moving beyond diagnostic labels, expanding disciplinary frameworks, and building humanistic responses; and (b) providing effective recovery-supportive responses that are guided by recovery-supportive perspectives, relationships, and multidisciplinary approaches. Lorien, Blandon, and Madsen [16] found the adoption of personal recovery as the guiding principle, and an application of a model of care that emphasizes relational care, culturally appropriate care planning, and psychosocial rehabilitation to be critical elements in the implementation of recovery-oriented practice in hospital-based mental health services. Winsper and colleagues [17], in their review of 309 papers based on a logic model, found psycho-education, peer support, social inclusion, and pro-recovery and mental health literacy training as the four main recovery-oriented types of intervention that have positive effects on functional recovery, existential recovery, and social recovery. 

The key principles and philosophies underlying various models of recovery-oriented practice include an emphasis on processes that foster (a) living a good life through promoting wellbeing, (b) an orientation towards the primacy of the person in determining priorities and aspirations, and (c) maintaining citizenship and social inclusion. For example, Slade [18] proposed a recovery-oriented practice that included the four key processes of hope, identity, meaning, and personal responsibility, based on the concept of lived experiences and positive psychology as the science of wellbeing. Further, Davidson and colleagues [19] developed their model of recovery-oriented practice guided by a concept of recovery that refers “primarily to a person diagnosed with a serious mental illness reclaiming his or her right to a safe, dignified, and personally meaningful and gratifying life in the community”, while also emphasizing self-determination (p. 11). To these authors, ‘living well’ meant living, working, learning, and participating fully in the community.

In embracing the prevailing ideas in the literature on recovery and recovery-oriented practice and integrating our orientation toward lived experience, we identified three general philosophical/conceptual ideas as the basis for an elaboration of our model of recovery and recovery-oriented practice. These include the principles of wellbeing, the principles of person-centeredness and self-determination, and the general orientation of human life as interdependent and social. 

## 3. The Principles of Wellbeing

Wellbeing has become a central focus of international health care policy, suggesting the need for a shift from a traditional deficit focus in mental health care to a focus on possibilities and the necessary conditions for living well [20]. This focus on wellbeing is in line with the WHO definition of mental health, suggesting that it is: “a state of wellbeing in which the individual realizes his or her own abilities, can cope with the normal stresses of life, can work productively and fruitfully, and is able to make a contribution to his or her community” [21]. A variety of theories have attempted to capture the key aspects of wellbeing. Historically, a main distinction has been made between two approaches, both drawn from Greek philosophy. First, the hedonic view is that wellbeing is the experience of pleasure and the minimization of pain. Gordon and Oades [22] argue that this view relates closely to having one’s needs satisfied and experiencing positive emotions such as joy and happiness. Alternatively, drawn from the work of Aristotle, the eudemonic view of wellbeing may be considered as flourishing, in which one is moving to reach one’s potential, in line with personal values [22,23]. Embedded in the concept of eudemonic wellbeing is the recognition that personal wellbeing cannot be separated from the wellbeing of others and, therefore, needs to encompass concern for others [24]. Current shorthand definitions of wellbeing often combine these two approaches in phrases such as “feeling good and functioning well” [25] (p. 8). 

The concept of wellbeing has also been extended in positive psychology, which focuses on factors that enable human flourishing at individual, community, and societal levels. Seligman [26] specifies five elements in his PERMA model (positive emotion, engagement, positive relationships, meaning, and accomplishments) as the basis for wellbeing. Slade [18] also adopts the tenets of positive psychology as the foundation for his conceptualization of personal recovery as focusing on wellbeing. The idea of recovery is couched in the philosophy of a good life and wellbeing, recognizing the individual’s efforts in striving to in live on his/her terms which bring about happiness/flourishing. Furthermore, happiness/flourishing is achieved through rationality, emotionality, coordination, and intersubjectivity in the contexts of the self and the environment. The significance of environmental and social aspects of wellbeing has been elaborated by Keyes [27], who points out that social wellbeing includes dimensions such as social integration, social contribution, social coherence, social growth, and social acceptance. 

### 3.1. The Principles of Person-Centeredness and Self-Determination

The notions of person-centeredness and self-determination have emerged strongly in the health care field in recent decades as a perspective in which the worthiness of individuals as autonomous, unique human beings is upheld as the foundation for a meaningful and fulfilling life. These notions are, therefore, viewed as the basis for wellbeing and health, and are founded upon the concept of personhood, which upholds the essential features of being human in terms of dignity, autonomy, and human singularity [28]. Person-centeredness, therefore, means that people (a) have the human dignity to be treated by self and others with respect and individuality, (b) have autonomy and self-determination regarding self and life, and (c) are unique individuals in terms of their human characteristics and their life experiences. Person-centeredness is, therefore, intrinsically connected to self-determination, which will ensure the exercise of individuality/singularity and individual autonomy. However, as the nature of human existence is continuously and complexly embedded with the lives of others and environmental factors, self-determination is morally constrained by the need for coordination in terms of justice and rightness in relation to self and others, and here social order plays a critical role [29]. Self-determination is, therefore, intrinsically framed by Aristotle’s virtue ethics, which state that what one decides to do for oneself and one’s own flourishing must also be good for others to be of universal excellence [23]. The nature of human life as complex and intertwined has been extended by Habermas [30] with his discourse ethics, in which a critical factor is mutual understanding among people through dialogue.

### 3.2. The Principles of Social Interdependence of Human Life

Beginning with the thoughts of the classical philosophers, the essential features of human life have been examined in relation to the social component, where politics was the major aspect in Plato and Aristotle. Sociologists in modern times, starting with Weber, Durkheim and Marx and continuing to the post-modernists of recent decades, have pointed out various systematic ways in which social systems and social factors determine and constrain human experiences and actions. In a fundamental way, human living is played out in social contexts in which other people and social institutions both promote and confine the ways in which life is lived. In our work, the work of Bourdieu on human practice and social capital provides a foundation to consider how social aspects of human life influence everyday living, i.e., human practice. In his conceptualization of human practice, Bourdieu posits the role of capital in various forms, including economic, cultural, social, and symbolic capital, as resources that yield power playing out in human practice [31]. To Bourdieu, all forms of capital are the resources individuals use in living their lives, more specifically defined as “human practice”. Bourdieu emphasizes how social capital is particularly vital in human practice, defining it as “the sum of the resources, actual or virtual, that accrue to an individual or a group by virtue of possessing a durable network of more or less institutionalized relationships of mutual acquaintance and recognition” [32]. In this conceptualization, social capital is intrinsically tied to the nature of the enduring and continuing social relations in which individuals engage. Human life is played out in social contexts that include various forms of sustained relations for individuals as social participants, which are their source of social capital. Therefore, social interdependence with its various forms, qualities, and complexities will determine a person’s access to the power, in the form of resources, that is inherent in social capital. Although social interdependence includes complex forms of involvement in society, ranging from friendships to various social roles including citizenship roles, the focus for the development of our model is the notion of social capital embedded within the social interdependence of human life. Furthermore, in the development of the model, social capital and social interdependence are also understood as key to accessing other forms of capital, such as material and economic resources. Human practice thus also involves interdependence with contextual and non-human factors [33].

## 4. The Model of Recovery and Recovery-Oriented Practice

This model has been developed by integrating and consolidating the results of three meta-syntheses by this research team of the empirical publications published during the past 25 years. The major themes reported are presented in Table 1, as these themes became the components that elaborated the model.

The five major themes describing recovery experiences are viewed as specifying recovery as the dynamics between the self and others and as the dynamics between the self and material contexts (one’s environment) [2]. Recovery experiences of being normal, respecting and accepting oneself, being in control, and recovery as intentional, involve the dynamics of the self in striving, engaging in, and managing everyday life. Here, the person determines his/her own experiences in the context of other people in social settings that are critical elements of interactions, exchanges, and influences that promote or modify how the person experiences life. The person with MHSU problems on the journey to recovery thus has to coordinate with, respond to, and extract support from others in relation to their presence, interactions, and resources in a personally dynamic fashion. These dynamics are expressed in the four themes of the meta-synthesis as the dynamics of the self and others. The fifth theme in experiences of recovery is “recovery as material and social”, which refers to the experiences of being in society as a social agent engaged in various aspects of social life and formal and informal social settings, including the role of a citizen in society. The relational dynamics the person has to navigate and control in recovery involve dealing with stigma, social exclusion and isolation, and a loss of social roles, which are often associated with having MHSU problems, in dynamic processes of coordinating one’s own perceptions, desires, definitions, and preferences within the social environment. This theme also refers to experiences of recovery in one’s relationship to the material aspects of living, such as adequate housing and income. Material resources have to be available and the person must have the ability to access them. In these dynamics, the person in recovery has to identify the availability and accessibility of needed resources. 

From our perspective, the experiences of recovery are grounded in the recovery processes extracted in the second meta-synthesis. Three major themes extracted here (i.e., recovery processes as step-wise, cyclical, and continuous; recovery as everyday experiences; and recovery as relational) are then structured into three meta-concepts of being, doing, and accessing [3]. Recovery as being encompasses the non-linear, step-wise, and cyclical nature of life itself that is punctuated, accentuated, and complicated in various ways for a person with MHSU problems. Recovery as being thus aligns with the flow of living that is both predictable and unpredictable and involves every dimension of one’s living. On the other hand, recovery as doing refers to performing the activities of everyday living. A person with MHSU problems will find it difficult and complicated to engage in everyday activities in order to fulfill their obligations while also satisfying their desires. Thus, a recovery process of doing often involves learning new, revised, or temporary ways of dealing with various requirements for living in the context of MHSU problems and using MHSU services. Recovery as accessing, identified as the theme of “recovery as relational”, refers to ways that a person can have relationships with others and gain access to the environment and its resources for support, co-existence, and life satisfaction. As people with MHSU problems tend to be constrained in their abilities and/or in opportunities to access relationships and resources, recovery as accessing often requires extraordinary efforts to be proactive. 

While the concepts specified for the experiences and processes of recovery refer to the phenomena in persons in recovery, the major themes identified in our third meta-synthesis express the aspects of recovery-oriented practice considered as vital characteristics of such practice [4]. These themes were delineated from statements on the elements of processes in recovery-oriented practice desired by clients, clients’ family members, and practitioners in our studies. Four major themes identified in this meta-synthesis were reconceptualized in line with the notion of recovery-oriented practice as focusing on accessing and using various forms of capital necessary for recovery in coordination with persons in recovery, and on supporting the persons in expanding and enriching these. These were conceptualized in relation to material capital; personal capital of identity, hope, and other personal resources; and social and relationship capital [4]. Thus recovery-oriented practice applies various modes of helping, supporting, collaborating, relating, promoting, generating, and mobilizing these types of capital for persons in MHSU services so that the capital becomes available, accessible, and usable in recovery. Material capital refers to economic and other material resources that are necessary for everyday life. On the other hand, personal capital of identity, hope, and other personal resources encompasses the critical resources of strength that are necessary for the recovery processes to work well. Social and relational capital refers to the resources that are inherently present in the social interdependence of persons in their social networks, which endure as well as expanding or contracting. People draw on the strength inherent in social and relational capital to meet their needs for support, help, validation, and change. 

The model of recovery and recovery-oriented practice emerges from these three sets of recovery-related experiences as dynamically integrated, interrelated, and coalescing, because the locus of recovery-oriented practice is in the experiences of recovery coexisting with the processes of recovery. This is depicted in Figure 1. 

By integrating the results of our research, the model is founded in the philosophy of wellbeing and the good life as a human right, the principles of person-centeredness and self-determination as an ideal mode of living, and the philosophy of the interdependence of human living as the essential feature of the lives of humans as social and ecological beings. The concept of recovery is oriented towards the idea that one must strive to live as well as possible and must have access to and use of resources. This means that a person with MHSU problems may need relational and material/environmental support. Therefore, recovery-oriented MHSU services are viewed as supporting, facilitating, and coordinating the efforts of persons with MHSU problems in their recovery journey. The model as depicted in Figure 1 is configured by the complex interplay among three key components: (a) the recovery processes that result in experiences of recovery; (b) the processes of the recovery-oriented practice which supports the recovery journey; and (c) social and material capital, which is both the context composed of the capital in which the recovery journey occurs, and the existing capital influencing and requiring mobilization for the individual on the recovery journey, as well as for MHSU services. 

The model is constructed with the conceptual categories as the major components, rather than making direct use of the themes found in the meta-syntheses. This was to enable a high level of conceptualization derived from using the major themes as experientially critical categories. Table 2 presents three components in the model and their descriptors, mostly drawn from the themes extracted in the three meta-syntheses. These descriptors are empirically based, and thus specify the types of experiences and behavioral strategies that express the features of the components. However, this list of descriptors of these three components is not exhaustive and needs to be elaborated further through research. These descriptors express experiential and behavioral ways in which the model can be illustrated and understood.

This is primarily a descriptive model that provides a comprehensive picture of various experiential phenomena that exist in persons in recovery in the context of recovery-oriented practice. The experiences of recovery are viewed as inherently embedded within the processes of recovery involved in the recovery journey, while recovery-oriented practice influences and interacts with the person’s experiences and processes of recovery. These three components of experiences are interconnected in MHSU service provision at any given time and in long-term contact with services. A person begins a recovery journey with the realization of the MHSU problem and has experiences that reflect the dynamics of the self in relationships with others and with the environment through the ongoing processes of being, doing, and accessing. When the person receives MHSU services, he/she gains help, support, and coordination in a collaborative framework for accessing, mobilizing, coordinating, and using various forms of capital, through relationships with professional care providers. In an ideal situation, such practice processes will be tailored to individuals’ specific needs and desires. Since the themes found in the meta-syntheses are experiential, they can be used as illustrative components in understanding and describing experiences of recovery and recovery-oriented practice. Furthermore, the model may also be used to illustrate and explore how experiences, processes and practices are interrelated and interdependent. It is also possible to use this model in an explanatory fashion, by examining individuals’ recovery experiences in relation to the co-existing recovery processes and the use of MHSU care practices. Since it is an experientially based model, it is critical to examine how experiences in various dimensions exist for persons in recovery and in relation to practice processes either at a given time or over a longer period.

## 5. Discussion and Closing Remarks

The descriptive and explanatory model developed in this paper, based on our three previous meta-syntheses that compiled experiential knowledge from an extensive number of studies on recovery experiences, recovery processes and recovery-oriented services, demonstrates the complex and comprehensive nature of recovery. Personal experiences and processes are entangled with material issues, and, to enable recovery, having or developing access to a variety of resources is key. The model can be considered a useful tool in enhancing knowledge and understanding of how recovery at a personal level is entangled with relational and material issues, and how relational and material resources are also important at the level of service provision. Being sensitive to how social determinants affect possibilities for recovery is decisive in developing working relationships and adequate services. The model may be used to enhance understanding at an individual-oriented level, at a relational level and at a more structural level, and how these levels interact and mutually affect each other. Thus, the model unfolds recovery as multifaceted and complex, extending beyond the often referred to and criticized understandings of recovery as individual journeys [9,19,34]. 

According to the WHO [1], recovery and the recovery-oriented approach in MHSU services and in broader communities predominantly focus on issues such as values, meaning and connection. This involves a holistic and broad understanding of MHSU issues, where helpful services involve much more than diagnosis, medication and providing individual treatment. It is vital to address the social determinants affecting people’s mental health, including relationships, education, employment, living conditions, community, spirituality, and artistic and intellectual pursuits (WHO [1] (p. 4)). However, as the model elucidates, an important aspect in addressing these issues is to support the person in accessing, mobilizing and utilizing personal, social and material resources. These resources may also be referred to as forms of recovery capital. Recovery capital is used as a concept to clarify the impact of a person’s environment. Tew [35] suggests that the concept of recovery capital can provide systematic assessments of a person’s resources and strengths and focus on what is needed to support the recovery process. According to Best and Laudet [36] (p. 6), recovery capital can be defined as the “sum of resources and supports available to people starting recovery journeys”. Recovery capital involves a variety of internal and external resources that facilitate recovery, and is commonly considered to deal with available economic, social, personal and relationship capital in addition to the kinds of capital that may be lacking [9,35]. Thus, as described in this paper, the concept of recovery capital demonstrates the interdependence of human lives and their dependence on access to relational and material resources. However, these resources are considered to be unequally distributed in society. Thus, social and structural inequities may be barriers to recovery by hindering access to recovery-supporting resources [33]. 

While what is referred to as recovery capital involves a myriad of resources, Best and Laudet [36] argue that the actual growth of recovery capital is personal and idiosyncratic, but its impact can also be measured in social communities. Despite acknowledgment of the personal aspect of recovery capital and the need to support and create access to these resources, as specified in the model, it is also necessary to emphasize that access to recovery capital is unequally distributed. This requires improvement of the distribution of and access to capital at the institutional and personal levels, especially in light of the principle that humans have rights to the resources and capital necessary for living as well as possible. There is, thus, a close connection between recovery capital and human rights. Connecting recovery capital and human rights involves recognizing access to rights as a crucial form of capital. This is in line with Davidson et al. [19], who state that in order to recover and to develop a sense of being a full citizen in the community, a person must have certain rights and resources. Thus, as elaborated in the model in this paper, help in accessing, coordinating, mobilizing and using personal, social and material capital also involves access to human rights.

International human rights provide an important framework, obliging countries to respect, protect and fulfil fundamental rights and freedoms for all people, including the rights of people with mental health conditions and psychosocial disabilities. A rights-based approach to mental health involves a paradigm shift in what counts as knowledge, and hence also a shift in power balances, which aligns well with an understanding of recovery as also involving a paradigm shift [37]. The UN Universal Declaration of Human Rights [38] stipulates a range of civil, cultural, economic, political and social rights. Economic, social and cultural rights include the right to health, housing, food, education, employment, social inclusion and cultural participation [1]. The close link between recovery and human rights has been described in several studies and guidelines, which recognize how social and economic conditions are key to recovery [1,8,34]. Drawing a close connection between recovery capital and human rights underscores how human rights are connected to actual lived experiences and resources fundamental to living as well as possible. This relationship also suggests that support in accessing resources is consistent with recovery capital; despite its arguably idiosyncratic and personal aspect, such support also needs to be recognized as linked to human rights. To combat the uneven distribution of recovery capital and improve access to the resources involved, defining these resources as rights may be an important step towards a rights-based and holistic approach to mental health. The current experience-based model of recovery and recovery-oriented practice provides an important illustration of how recovery experiences and processes are personal and contextual, while recovery and recovery-oriented services may also be perceived as rights.

Mahdanian et al. [39] suggest the need for a more comprehensive model of mental health that integrates human rights into existing services and approaches and recognizes people with mental health conditions and psychosocial disabilities as rights holders. While not proposing the model in this paper as a comprehensive, all-encompassing model, we suggest that it provides useful possibilities for describing and exploring recovery and recovery-oriented services as being deeply imbedded in personal and idiosyncratic experiences and in a structural, human-rights based understanding of mental health and recovery.

## Figures and Tables

**Figure 1 ijerph-20-06607-f001:**
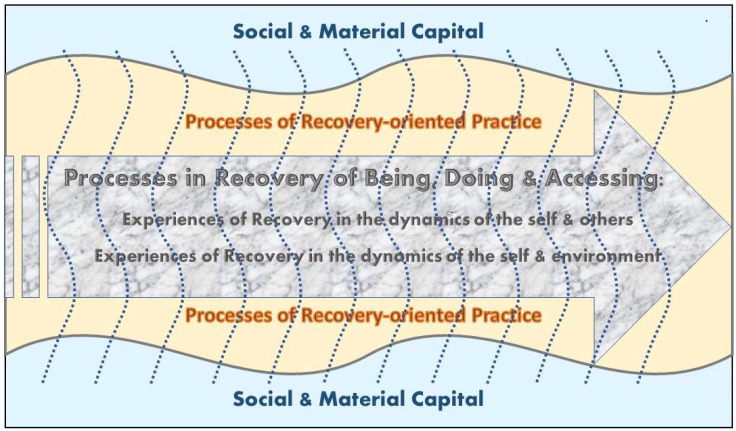
The Model of Recovery and Recovery-oriented Practice.

**Table 1 ijerph-20-06607-t001:** The major themes for recovery experiences, the processes of recovery, and recovery-oriented practice * (* these were reported in [2,3,4]).

The Experiential Contexts	Major Themes and Sub-Themes
**Recovery Experiences**	Experiences of being normal Respecting and accepting oneself Being in control Recovery experienced as intentional Recovery experienced as material and social
**Processes of Recovery**	Recovery processes as step-wise, cyclical, and continuous ■ Process involving steps forward and steps backward ■ Process involving all aspects of one’s life Recovery as everyday experiences ■ Struggling to achieve or remain in a normal, ordinary life ■ Accessing resources, possibilities, and enjoyment Recovery as relational ■ Developing and maintaining supportive relationships ■ Accessing supportive environments ■ Engaging in relational hope
**Recovery-oriented Practice**	Helping and supporting ■ Being helped on one’s own terms ■ Timely helping ■ Creative and collaborative helping and supporting ■ Applying helpful actions ■ Helping for different needs Collaborating and relating ■ Relational characteristics ■ Characteristics of professionals in collaborative relationships ■ Organizational conditions and strategies Identity integration in practice ■ Promoting individual identity ■ Promoting strength-oriented identity Generating hope through nurturing and helping ■ Supporting service users to become hopeful ■ Generating hope in the context of difficulties

**Table 2 ijerph-20-06607-t002:** Three components of the model of recovery and recovery-oriented practice and their descriptors.

Components of the Model	Modes of the Component	Themes
Recovery Experiences	Dynamics of the self and others	Experiences of being normal
		Respecting and accepting oneself
		Being in control
		Recovery experienced as intentional
	Dynamics of the self and environment	Recovery experienced in social relationships
		Recovery experienced in relation to one’s environment
Processes of Recovery	Processes of being	Step-wise, cyclical, and continuous processes. Processes involving steps forward and steps backward
		Processes involving all aspects of one’s life
	Processes of doing	Struggling to achieve or remain in a normal, ordinary life
		Seeking and utilizing resources in working toward possibilities and enjoyable aspects of life
	Processes of accessing	Developing and maintaining supportive relationships
		Accessing supportive environments
		Engaging in relational hope
Recovery-oriented Practice	General helping and supporting	Helping on one’s own terms
		Timely helping, helping for different needs, and applying helpful actions
		Collaborating and relating with creativity and applying various strategies
	Helping to develop and use personal resources	Helping to strengthen individual identity and promoting strength-based identity
		Helping to become hopeful by nurturing and supporting. Generating hope in the context of difficulties
	Helping in accessing, coordinating, mobilizing, and using social capital	Helping to maintain general social inclusion
		Helping to develop social relationships
		Helping to participate in ordinary social activities and in ordinary citizenship roles
	Helping in accessing, mobilizing, and utilizing material capital	Helping through coordination for accessing, mobilizing, and using necessary material resources for everyday life

## Data Availability

All the included studies that the model is based on can be found in references [2,3,4].

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
