# Peer review of "Toward an Experience-Based Model of Recovery and Recovery-Oriented Practice in Mental Health and Substance Use Care: An Integration of the Findings from a Set of Meta-Syntheses"

_ijerph, 2023, doi:10.3390/ijerph20166607_

Round 1
Reviewer 1 Report
Toward an Experience-Based Model of Recovery and Recovery-Oriented Practice in Mental Health and Substance Use Care: An Integration of the Findings From A Set of Meta-Syntheses
In this paper, the authors describe a model of recovery and recovery-oriented practice based on three previously published meta-syntheses of experiences and processes of mental health and substance use recovery. The experience-based model of recovery and recovery-oriented practice described by the authors shows how recovery processes are personal, while recovery-oriented services may also be perceived as rights.
Comments:
The manuscript analyses current and interesting topic. This paper is the first to integrate the findings of three previously published meta-syntheses of experiences and processes of mental health and substance use recovery. English language and style are fine. The conclusions are consistent with the evidence and the references are appropriate and exhaustive. The two tables and the only figures provide detailed information. The paper can be accepted after the authors have checked the format of all references according to the instructions of “IJERPH”. The paper can be accepted after minor revision.
Author Response
Thank you for your supportive feedback. In the revised and resubmitted paper, we have checked and edited the format of all references according to the instructions of “IJERPH".
Reviewer 2 Report
Thank you for the chance to review your manuscript on “Toward an experience-based model of recovery and recovery-oriented practice in mental health and substance use care: An integration of the findings from a set of meta-syntheses".
1. Page 2 of 16
"This paper presents an experience-based model of recovery and recovery-oriented practice developed by integrating the findings from three meta-syntheses (Klevan et al., 2021, Sommer et al., 2021, & Klevan, Sommer, Borg, Karlsson, Sundet, & Kim, 2021) that were based on the empirical publications of the research team of the Center for Mental Health and Substance Abuse of the University of South-Eastern Norway."
The authors may want to consider adding a clear explanation of their inclusion and exclusion criteria for selecting studies. This should include an explanation of why they chose the three studies included in the meta-synthesis, as well as a detailed description of the search criteria used to identify relevant studies. It is important for authors to clearly define the selection criteria for the meta-synthesis, including the time frame of the search and any specific search terms or keywords used. By providing a clear and detailed explanation of their selection criteria, the authors can help readers understand the rationale behind their choices and ensure that the meta-synthesis is based on a comprehensive and rigorous review of the literature.
2. Page 10 of 16
Figure 1
Authors may want to improve or reconstruct the Figure 1 to make it more clear in terms of of Recovery and Recovery-oriented Practice.
3. The current study has a very limited sample size of only three studies, which may make it challenging to conduct a robust meta-synthesis. Therefore, the authors may want to consider expanding their research to include more studies in order to construct a more comprehensive model of Recovery and Recovery-oriented Practice. This would allow for a more thorough analysis of the research question and provide a more robust and reliable set of findings. By including a larger number of studies in the meta-synthesis, the authors can increase the validity and generalizability of their results, and ensure that their conclusions are based on a more relevant literature.
Quality of English Language is acceptable.
Author Response
1. Thank you for your feedback. The paper is not a systematic review paper, but a paper coalescing the results from our three meta-syntheses papers. These three papers are referenced in the current paper. These three meta syntheses are, put together, based on the findings of 74 papers (and not three). This could have been made clearer in the current paper and we have added information about this in the introduction in the revised version.
2. The reviewer has not written any comments about what is unclear about figure 1 and it is thus difficult to understand what changes the reviewer thinks should be made. We have chosen not to make changes to the model, as the model is explained in the text and other reviewers are satisfied with it.
3. See nr 1.
Reviewer 3 Report
Overall I found the paper very. interesting and well crafted. The authors made it clear why their study was needed, for example, in linking recovery practice to a human rights approach.
The paper appears useful in summarizing the authors’ own extensive research, and in summarizing other relevant literature.
The conclusions were logical and well supported.
I did not see a discussion of potential limitations, though perhaps that relates to the methodology. The conclusions were reasonable as were the suggestions for further research
In Section
3.2, 6th line: the sentence reads “...social systems and social factor determine …”. Unless there is always exactly one factor involved, it might be better to use the plural form of the word: “factors”. If “factor”is intended, then the sentence needs to be reworded in order to be grammatically correct and not sound awkward.
--> I have a question about the term “emerging in relational hope”, which appears in Table 1 and also in Table 2. I don’t remember seeing this discussed anywhere in the paper, though I might have missed it. That could be confusing to readers who are not familiar with the concept.
Please see above. Minor corrections are needed.
Author Response
Thank you for your feedback.
In 3.2, line 6, we have changed factor to factors.
Regarding the term "emerging in relational hope" - this should be "engaging in relational hope". This is a theme from one of the previously published meta-syntheses. We have changed this in the paper. Thank you for noticing this.
Reviewer 4 Report
Dear author,
plagiarism percentage in very high. Before review the manuscript, author first need to remove the plag and then resubmit accordingly.
English very difficult to understand/incomprehensible
Author Response
We find that the comment about plagiarism by Reviewer #4 is unacceptable, this comment is neither understandable nor addressable.
Neither is the comment about language understandable. The other three reviewers had no comments about language and the paper has been proofread and edited by a professional proofreader. Without any further information or examples, sweeping comments like these are of no use and we have not addressed them.
Round 2
Reviewer 2 Report
All comments are appropriately addressed.
Author Response
Thank you.
Reviewer 4 Report
Dear Author,
Same comments: After carefully applying of official plagiarism software, it was found that plagiarism percentage in very high.
Before giving any comment, and review the manuscript, author first need to remove the plagiarism and then resubmit accordingly.
Following section need extensive removal of plagiarism:
1. Background and foundations
2. Principles and wellbeing
3. Table 1 is 99% plagged.
4. Table 2 need extensive plag removal
5. Discussion and closing remarks.
English very difficult to understand/incomprehensible
Author Response
We find that we cannot revise the paper responding to the Reviewer #4’s remarks and would like to give the following response:
1. The reviewer indicates that the plagiarism claim is based on the result of the manuscript review applying the Official Plagiarism Software. This result is understandable as it is due to the nature of this manuscript which is the consolidation and analytical systematization of the three previous meta-synthesis papers by the same group as authors (us). We certainly have used the same or similar wording, terms, and sentences in this manuscript because we were analytically integrating the results of these three papers that we have written, including using the same tables that were in the three papers. We believe that one cannot plagiarize one’s own work as the work belong to the same author! We think the reviewer should have reviewed the results of this outcome from the application of the Plagiarism App more responsibly by investigating the results further to tease out this type of misrepresentation.
2. In addition to the work being very closely based on, and bringing together, our own previous work published as a series of papers in IJERPH, we also reference some other important, conceptual papers on recovery and their key elements. For example, we mention the recovery framework CHIME and describe its elements. This might also appear as plagiarism, but referencing papers like this and reporting their key elements involves that many of the same words are used. We have cited the work of others every time we reference somebody else's work. We also cite our own, previous work that the current paper brings together all the way throughout the paper.
2. We have had the manuscript reviewed for its quality and correctness regarding the English language by a professional service that we have used numerous times, and we were assured by the three other qualified reviewers that the manuscript is in a satisfactory, acceptable state.